# DMTD: DYNAMIC MULTI-TEMPERATURE DISTILLATION

## ABSTRACT

Knowledge Distillation serves to compress complex neural networks into simpler architectures. In recent years, logit-based distillation has gained significant attention due to its computational efficiency. However, many logit distillation techniques merely set a single temperature as a fixed parameter, which is only sensitive to global features. In contrast, numerous computer vision tasks require varying degrees of attention to both global and local features simultaneously. This contradiction hinders the effectiveness of logit distillation methods in computer vision applications. To this end, this paper introduces a modular approach known as **D**ynamic **M**ulti-**T**emperature **D**istillation (**DMTD**), which employs multiple learnable temperatures by adaptively adjusting temperature parameters based on the significance of both global and local features. This method enhances the student model's ability to mimic the hidden behaviors of the teacher during inference. Experimental results demonstrate that DMTD integrates effectively with existing logit distillation methods, leading to significant improvements across various teacher-student pairs in benchmarks for image classification and object detection.

## 1 INTRODUCTION

In recent years, Knowledge Distillation (KD) has garnered significant attention in computer vision field (Zhang et al., 2024; Liu et al., 2019; Wang et al., 2024) for its effectiveness in compressing deep networks to accommodate edge devices with limited computational resources. It is a general technique that leverages the knowledge from a pre-trained teacher network to enhance the performance of a student model while maintaining its lightweight network scale (Gou et al., 2021). The concept of KD was first introduced in (Hinton, 2015), where knowledge is transferred by minimizing the Kullback-Leibler ($KL$) divergence between the probability distributions predicted by the teacher and student networks, shown in Fig. 1.

From FitNet (Romero et al., 2014), considerable efforts have been devoted to the simultaneous extraction of knowledge from deep features at intermediate layers. Since then current distillation methods can be categorized into three main types: feature-based distillation (Chen et al., 2021; Wang et al., 2019), logit-based distillation (Zhao et al., 2022; Yang et al., 2023) and a combination thereof (Huang et al., 2022). Feature-based distillation has the potential to convey richer knowledge to the student by facilitating the comparison and alignment of features at various layers between the teacher-student pair, thereby enhancing the student's understanding of intrinsic structure of the input. However, this manner results in increased training costs and strict requirements of the architectural characteristics of the teacher-student models which significantly influence its effectiveness. Logit-based methods, in contrast, exclusively necessitate knowledge distillation at the deepest semantic layer. This character endows them with inherent computational efficiency (Zhao et al., 2022) and the ability to manage heterogeneous knowledge (Wei et al., 2023). Hence, these methods have obtained significant attention from researchers aiming to further enhance their performance.

As shown in Fig.1, the standard procedure for logit distillation involves minimizing the $KL$ divergence loss between the prediction of the teacher and that of the student, as illustrated in Eq. (1):

$$Loss_{KD} = \tau^2 KL \left( \sigma \left( \frac{q_t}{\tau} \right), \sigma \left( \frac{q_s}{\tau} \right) \right) \qquad (1)$$

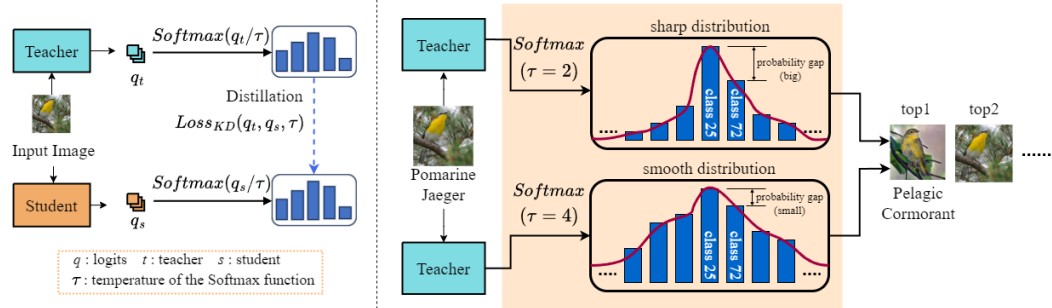

Figure 1: **left:** Conventional KD for image classification on CUB200 (Welinder et al., 2010). **right:** The probability distributions when the temperature $\tau$ setting to different values.

where $q_t$ and $q_s$ represent the logits produced by the teacher and student, respectively, $\sigma\left(\cdot\right)$ is the $Softmax$ function, and $\tau$ serves as a key temperature hyperparameter to regulate the smoothness of the predictive probability distributions generated by both the teacher and the student (Hinton, 2015).

Take a classification task as example, shown in Fig. 1, a lower temperature ($\tau = 2$) results in a sharper output probability distribution (Hinton, 2015), and the prediction exhibits a high level of confidence regardless of whether the predictions are correct or not, the top1 classification result shows a high probability, while there is a significant probability-gap between the classes of top2 and top1. In this case, the student model will primarily focus on the class with the highest predicted probability from the teacher, while under-emphasizing relationships among other classes. In scenarios where the teacher model can accurately classify certain classes, employing a low temperature is evidently advantageous. In contrast, a higher temperature smooths the output distribution, placing greater emphasis on global features, thereby maintaining the ambiguity of samples. When $\tau = 4$, the bird of Class-72 is misclassified as Class-25 in the top1 prediction, while the top2 classification is correct, it is evident that a small probability-gap would be preferable. Because, in scenarios where the teacher fails to classify correctly, it helps to reduce the probability-gap among different classes, which allows the student model to have opportunities for error correction when faced with ambiguous or challenging samples, ultimately enhancing its own accuracy.

Consequently, it is crucial to determine an appropriate temperature, and there has already been some work dedicated to this issue, as evidenced by references (Chen et al., 2021; Wei et al., 2023; Ji et al., 2021). However, a fixed temperature may not always be optimal, because it is too rigid for the student to align with the teacher's probability distribution across different samples. To solve this issue, Meta Knowledge Distillation (MKD) (Liu et al., 2022) proposes the learning of a dynamic temperature through meta-learning, its primary idea is to utilize an additional validation set to train the temperature module. Another approach, such as KD-fn (Xu et al., 2020), uses a fixed temperature parameter $\tau$ combined with the $L_2$ norm of the penultimate layer features to replace the uniform temperature $T$, allowing sample-specific noise handling.

The introduction of these methods contributes to determining a reasonable temperature value for the logit distillation. However, they still present many disadvantages. For instance, the MKD method necessitates an additional validation set to train the temperature module, thereby complicating the training process. Similarly, the KD-fn method necessitates extra mean calculations. Moreover, these methods may yield sub-optimal results as they apply merely one global temperature across all images while neglecting detailed information contained within local features, thereby impeding the potential performance enhancements. At last, they can hardly be seamlessly integrated with existing advanced logit distillation techniques.

In response to the aforementioned issues, we propose a novel plug-in method that enables the temperature to be multiple and learnable, referred to as Dynamic Multi-Temperature Distillation (DMTD). Specifically, through DMTD, the feature maps are replicated into multiple sets, with each set further divided into different numbers of patches. Average pooling operations are performed on the local features within each set, generating multiple sets of logit outputs. Subsequently, for each set of logits, a $Softmax$ layer with different temperature parameters is configured to obtain the corresponding probability distributions. It is worth noting that all local logit outputs within the same

set share the same temperature parameter, which is determined by a learnable temperature module. Finally, the performance of the student model is enhanced through by conducting one-to-one knowledge distillation using the $KL$ divergence loss function between the probability distributions predicted by the teacher and the student. Overall, our contributions can be summarized as follows:

- We proposed a plug-in method based on multiple learnable temperatures that processes logits obtained from pooled feature maps at various scales through $Softmax$ layers with different temperature parameters and then performs one-to-one knowledge distillation to optimize the performance of the student model.

- We first integrated the dynamic multi-temperature allocation module into multi-scale logits distillation, addressing the limitation of a single temperature parameter in traditional knowledge distillation methods, fully leveraging the advantages of features at different scales, and significantly enhancing the performance of the student model.

- We conducted a lot of experiments, which demonstrate that the proposed Dynamic Multi-Temperature Distillation strategy can be conveniently combined with representative logit distillation methods and prove the effectiveness of DMTD for a wide range of teacher-student pairs on several image classification and object detection benchmark datasets.

## 2 RELATED WORK

### 2.1 KNOWLEDGE DISTILLATION

Knowledge distillation (KD) research can be categorized into two main types: extracting knowledge from logits and extracting knowledge from intermediate features. The former focuses on transferring knowledge through the output logits, with seminal work like KD (Hinton, 2015) minimizing the $KL$ divergence between the teacher and student predictions. Methods like Decoupled Knowledge Distillation (DKD) (Zhao et al., 2022) decompose the loss into target and non-target class components, improving efficiency for well-predicted samples, while Normalized Knowledge Distillation (NKD) (Yang et al., 2023) normalizes non-target logits to enhance student learning. The latter category, exemplified by FitNet (Romero et al., 2014), transfers knowledge via intermediate features by aligning teacher-student representations across layers and minimizing MSE loss. CRD (Tian et al., 2019) captures structured knowledge by using contrastive objectives to model higher-order dependencies between features. MGD (Yang et al., 2022) enforces feature completeness by masking random pixels in student features and forcing the student to reconstruct the teacher's complete features. SP (Tung & Mori, 2019) preserves activation similarities by ensuring that input pairs with similar activations in the teacher network produce similar activations in the student network. SemCKD (Wang et al., 2022) enables flexible cross-layer supervision by using attention mechanisms to assign appropriate teacher layers to each student layer, allowing each student layer to extract knowledge from multiple teacher layers. Additionally, theoretical works like (Phuong & Lampert, 2019) and (Stanton et al., 2021) have provided insights into the mechanisms underlying KD's effectiveness.

### 2.2 WORKS WITH TEMPERATURE

The importance of temperature tuning in deep learning models has been widely recognized. In contrastive learning (Sun & Li) and reinforcement learning (Asadi & Littman, 2017), adaptive temperature selection during training has been shown to optimize strategies. In knowledge distillation, KD-fn (Xu et al., 2020) uses a fixed temperature parameter $\tau$ combined with the $L_2$ norm of the penultimate layer features to replace the uniform temperature $T$, allowing sample-specific noise handling. Meanwhile, MKD (Liu et al., 2022) meta-learns a learnable temperature by minimizing validation loss during student training.

However, both methods employ a single global temperature, neglecting the knowledge of the local features. In this work, we propose multiple learnable temperatures for the first time, which process logits from feature maps at various scales through softmax layers with different temperatures and perform one-to-one knowledge distillation to optimize performance. Furthermore, DMTD operates under normal augmentation, thus it can be seamlessly integrated into existing KD frameworks.

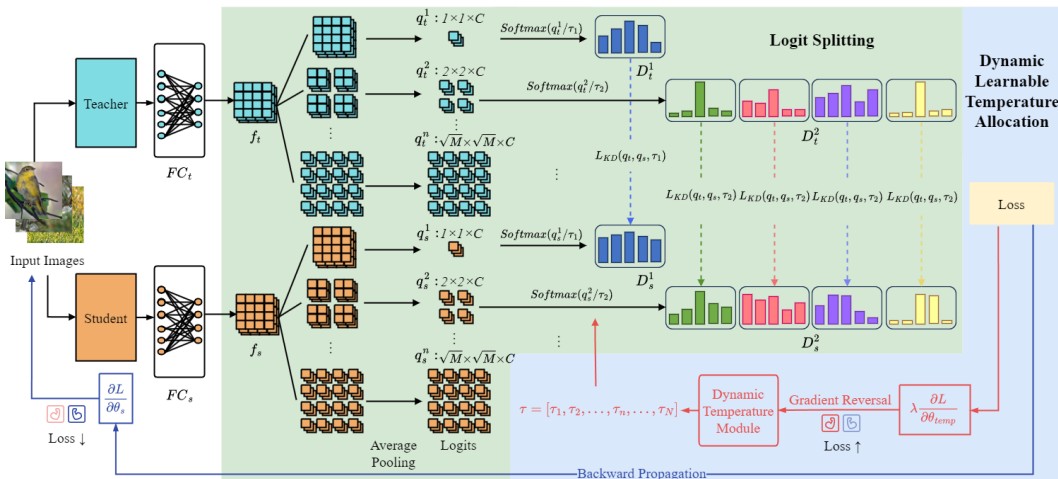

Figure 2: Framework diagram of the proposed Dynamic Multi-Temperature Distillation (DMTD). The **logit splitting module** splits feature maps into multiple sets with varying numbers of patches and generates logit outputs at different scales via average pooling, and the **dynamic learnable temperature allocation module** configures softmax layers with different temperatures for each set of logits, adaptively adjusted by a learnable module to optimize distillation.

## 3 METHODOLOGY

### 3.1 OVERVIEW OF DMTD

In this part, we will describe in detail the proposed Dynamic Multi-Temperature Distillation (DMTD). As shown in Fig. 2, DMTD consists of two parts: (1) Logit Splitting Module. The primary objective of this module is to address the limitations inherent in conventional KD, where a single global temperature may impede the model's ability to fully leverage semantic information derived from local features. By partitioning the final feature maps of both the teacher and student models into varying numbers of patches and generating multiple sets of logit outputs across different scales, this approach lays the groundwork for dynamic multi-temperature allocation. This enables the student model to more effectively emulate the behaviors of the teacher by utilizing both global and local features at diverse scales, thereby enhancing overall distillation performance. (2) Dynamic Learnable Temperature Allocation Module. To enable the student model to dynamically adapt its focus among global and local features, this well-designed module facilitates the adaptive adjustment of the temperatures for each output logits during training when optimizing the distillation loss. This will enhancing the student's ability to mimic the teacher model's behavior more effectively.

### 3.2 LOGIT SPLITTING

Suppose the input image is $x$, and its corresponding label as $y$ with $(x, y) \in D$, where $D$ represents a specific dataset. After each fully connected layer of the teacher and the student, the feature maps of them are represented as $f_t(x)$ and $f_s(x)$.

Both feature maps are first duplicated as $N$ sets and each set is then splitted into a varying number of patches, where the varying number is denoted as $M \in \{z^2 \mid z \in \mathrm{Z}^+\}$. For the teacher and student separately, these sets of duplicated feature maps can be represented as:

$$\{f_t^n(x)\}_{n=1}^N, \{f_s^n(x)\}_{n=1}^N \tag{2}$$

where $n$ indicates the $n$-th set. Obviously, any two set of feature maps in $\{f_t^n\}_{n=1}^N$ or in $\{f_s^n\}_{n=1}^N$ are identical essentially but different on scales. Each set of feature maps can be regarded as a complete feature map composed of multiple local feature maps. For each $f_t^n, f_s^n$, their local feature maps are:

$$f_t^n = \left[f_t^{n,1}, f_t^{n,2}, \ldots, f_t^{n,M}\right], f_s^n = \left[f_s^{n,1}, f_s^{n,2}, \ldots, f_s^{n,M}\right] \tag{3}$$

$M$ represents the number of split local feature maps for each set, all values of $M$ are the square of the element $z$ in the positive integer set $Z^+$.

For each local feature map in the $n$-th set ($n \in N$), average pooling operations are performed to obtain the $n$-th set of local logits $q_t^n$ or $q_s^n$:

$$q_t^n = \text{AvgPooling}(f_t^n) = \left[ q_t^{n,1}, q_t^{n,2}, \ldots, q_t^{n,M} \right],$$
$$q_s^n = \text{AvgPooling}(f_s^n) = \left[ q_s^{n,1}, q_s^{n,2}, \ldots, q_s^{n,M} \right] \tag{4}$$

### 3.3 DYNAMIC LEARNABLE TEMPERATURE ALLOCATION

For each set of logit outputs, a $Softmax$ function (denoted as $\sigma(\cdot)$) with different temperature parameters $\tau_n$ is configured to obtain the corresponding probability distributions $D_t^n$ and $D_t^n$:

$$D_t^n = \text{Softmax}(q_t^n/\tau_n) = \left[ d_t^{n,1}, d_t^{n,2}, \cdots, d_t^{n,M} \right],$$
$$D_s^n = \text{Softmax}(q_s^n/\tau_n) = \left[ d_s^{n,1}, d_s^{n,2}, \cdots, d_s^{n,M} \right] \tag{5}$$

and for each set, the number of probability distributions converted from logit outputs is equal to the number of local feature maps in this set. It is important to note that all local logit outputs within the same set share a common temperature parameter to ensure the comparability of the probability distributions between the one-on-one teacher-student pair. This parameter is computed by a dynamic temperature module, which adaptively learns and adjusts the subsequent multi-temperature set:

$$\tau = [\tau_1, \tau_2, \cdots, \tau_n, \cdots \tau_N] \tag{6}$$

In the training process, specifically, the temperature parameters are trained following the idea of curriculum learning (Bengio et al., 2009) (Li et al., 2023), and the adversarial role increases by combining reversed gradients of $\tau_n$ with a weight $\lambda$, which is continuously increased (e.g., following a cosine curve from 0 to 1 over 10 epochs and then kept constant). This adaptive learning process allows the student model to gradually leverage knowledge from both global and local features, enhancing its ability to mimic the teacher model's behavior more effectively.

**Distillation Loss.** The student model is optimized by conducting one-to-one knowledge distillation using Eq. (1) between all corresponding probability distributions predicted by the teacher model and the student model. Here, to further enhance the distillation performance, we introduce a knowledge weighting operation. Specifically, the logit outputs are divided into consistent terms and inconsistent terms based on whether the teacher makes correct inferences. When the ground truth is consistent with the teacher's predicted logits, the local logits can be regarded as a supplement to the ground truth. Conversely, when the ground truth is inconsistent with the teacher's predicted logits (i.e., the teacher's local logits make a wrong inference), it indicates ambiguity in the sample. According to SDD (Luo, 2024), the wrong inference results obtained by the teacher based on certain local logits are equally important, as preserving sample ambiguity can avoid overfitting results of the student. Additionally, the distribution of incorrect predictions may implicitly contain the teacher model's understanding of sample similarity and the spatial relationships between internal elements of the images, which may be difficult for the student model to learn on its own. Therefore, we introduce an independent hyperparameter $\beta$ to weight the losses incurred by the student when learning these two types of knowledge. This knowledge weighting operation achieves better distillation performance by retaining the ambiguity of samples. Hence, by adding up each $KL$ divergence, we can convert the Eq. (1) to our DMTD loss as follows:

$$L_{DMTD} = \sum_{n=1}^{N} \sum_{m=1}^{M} \tau_n^2 \cdot w \cdot KL\left( \sigma\left(\frac{q_t^{n,m}}{\tau_n}\right), \sigma\left(\frac{q_s^{n,m}}{\tau_n}\right) \right) \tag{7}$$

where $w$ represents the weighting coefficient. When the prediction results generated from $q_t^{n,m}$ and the ground truth are consistent, $w$ is set to 1; and when they are inconsistent, $w$ is set to $\beta$. Unless otherwise specified, $\beta$ is defaulted to 2 in all subsequent experiments throughout this paper.

**Total Loss.** Combining the distillation loss defined above, the total training loss is defined as:

$$L = L_{CE} + \alpha L_{DMTD} \tag{8}$$

where $L_{CE}$ is the label supervision loss for the current task, and $\alpha$ is a balancing factor to ensure the student to make reasonable and full use of the global and local knowledge provided by the teacher.

## 4 EXPERIMENTS

### 4.1 EXPERIMENTAL SETTINGS

**Datasets.** We conduct classification experiments on CIFAR-100, Tiny-ImageNet (Krizhevsky et al., 2009), and CUB200 (Welinder et al., 2010). Here, CIFAR-100 and Tiny-ImageNet are used for evaluating classic classification tasks. CUB200 is used to assess fine-grained classification tasks, which include 200 different bird classes.

We perform detection experiments on MS-COCO (Lin et al., 2014), an 80-class general object detection dataset. The train2017 split contains 118k images, and the val2017 split contains 5k images.

**Implementation Details.** As can be seen from Eq. (7), any KD method that employs a logit-based loss is compatible with our plug-in DMTD. Here, to ensure a fair comparison with logit-based methods and to investigate the effect of DMTD, we use the same losses as in KD, DKD, and NKD methods, and refer to these implementations as DMT-KD, DMT-DKD, and DMT-NKD, respectively. In addition, we compared these three methods with RLD (Sun et al., 2024) on CIFAR-100.

DMTD has three hyperparameters: the number of split local feature maps $M$, and the balancing parameters $\alpha$ and $\beta$. Empirically, for distillation tasks with different architectures, we set $M$ to 1, 4, and 16, while for those with similar architectures, we set $M$ to 1 and 4. $\beta$ is set to 2. As for $\alpha$, to conduct a fair comparison with previous methods, we follow the same settings used in the original KD, DKD, and NKD. It is worth noting that due to the potential high summation of the DMTD loss, which may result in a large initial loss, we employ a 30-epoch linear warm-up for all experiments.

**Training Details.** For CIFAR-100 and CUB200, our implementation follows the practices in CRD (Tian et al., 2019). The teacher and student are trained for 240 epochs with SGD. The batch size is set to 64, and the learning rates for ShuffleNet and MobileNet-V2 are 0.01, while for other series (e.g., VGG, ResNet, and WRN), it is 0.05. The learning rate is divided by 10 at epochs 150, 180, and 210. Weight decay and momentum are set to 5e-4 and 0.9, respectively. For a fair comparison, the weights of the distillation loss follow the same settings as in KD, DKD, and NKD. For Tiny-ImageNet, we follow the settings in (Yuan et al., 2020). All models are trained for 200 epochs with the learning rate decayed by 0.1 at epochs 60, 120, and 160. For a batch size of 64, the initial learning rate is set to 0.05. We use SGD with 0.9 momentum and 0.0005 weight decay as the optimizer, the weights of the distillation loss follow the same weights as in KD, DKD, and NKD.

Our implementation for MS-COCO follows the settings in (Chen et al., 2021). We use a two-stage method Faster R-CNN (Ren, 2015) with FPN (Lin et al., 2017) as the feature extractor. ResNet-101 and ResNet-50 are selected as teachers, while ResNet18 and MobileNet-V2 (Sandler et al., 2018) are chosen as their students, respectively. The student is trained with 1×scheduler (scheduler and task-specific loss weights follow Detectron2 (Wu et al., 2019)). We adopt the DMT-DKD loss at the R-CNN head.

### 4.2 MAIN RESULTS

**CIFAR-100 Classification.** Tab. 1 shows the top-1 classification accuracy on CIFAR-100 based on 6 different teacher-student pairs. We can observe that all different student networks benefit from our method, with improvements being quite significant in some cases.

**Tiny-ImageNet Classification.** Tiny-ImageNet is a more challenging dataset compared to CIFAR-100. The results in Tab. 2 demonstrate that our DMTD still achieves more significant performance improvements on such a challenging dataset. DMT-DKD raises the top-1 accuracy of the ResNet50-MobileNetV1 teacher-student pair from 71.15% to 73.12%, and DMT-KD increases the top-1 accuracy of the ResNet34-ResNet18 pair from 71.52% to 72.33%. Notably, the performance of DMT-NKD surpasses the results of SOTA logit distillation methods. The results indicate that DMTD can still effectively handle large-scale datasets.

**CUB200 Classification.** As shown in Tab. 3, DMTD improves the performance of various classic logit distillation methods by 0.25%-8.03%. This notable improvement is particularly evident in fine-grained tasks like CUB200, where inter-class variations are subtle. These results demonstrate that the proposed DMTD achieves even more significant performance gains on such a task where different classes have small differences. The reason may be that different classes share similar global

Table 1: Top-1 accuracy of student networks on CIFAR-100.

| Distillation Manner | Teacher | ResNet32×4 | WRN40-2 | ResNet50 | VGG13 | ResNet32×4 | ResNet32×4 |
|---|---|---|---|---|---|---|---|
| | Acc | 79.42 | 75.61 | 79.34 | 74.64 | 79.42 | 79.42 |
| | Student | ShuffleNetV1 | ShuffleNetV1 | MobileNetV2 | MobileNetV2 | ShuffleNetV2 | MobileNetV2 |
| | Acc | 70.50 | 70.50 | 64.60 | 64.60 | 71.82 | 64.60 |
| Feature-Based | FitNet (ICLR'15) | 73.54 | 73.73 | 63.16 | 63.16 | 73.54 | 65.61 |
| | SP (ICCV'19) | 73.48 | 74.52 | 68.08 | 68.08 | 74.56 | 67.52 |
| | CRD (ICLR'19) | 75.11 | 76.05 | 69.11 | 69.11 | 75.65 | 69.13 |
| | SemCKD (AAAI'21) | 76.31 | 76.06 | 68.69 | 69.98 | 77.02 | 68.99 |
| | MGD (ECCV'22) | 76.22 | 75.89 | 68.54 | 69.44 | 76.65 | 68.13 |
| Logit-Based | KD (NIPS'14) | 74.07 | 74.83 | 67.35 | 67.37 | 74.45 | 67.72 |
| | DKD (CVPR'22) | 76.45 | 76.70 | 70.35 | 69.71 | 77.07 | 68.98 |
| | NKD (ICCV'23) | 75.31 | 75.96 | 69.39 | 68.72 | 76.26 | 68.81 |
| | RLD (ICCV'25) | 76.93 | 77.12 | 70.76 | 69.97 | 76.58 | 69.05 |
| | DMT-KD (ours) | 75.13(+1.06) | 76.08(+1.25) | 68.57(+1.22) | 68.64(+1.27) | 75.90(+1.45) | 69.02(+1.30) |
| | DMT-DKD (ours) | **77.68(+1.23)** | **77.32(+0.62)** | **71.86(+1.51)** | **70.10(+0.39)** | **77.94(+0.87)** | **69.90(+0.92)** |
| | DMT-NKD (ours) | 76.43(+1.12) | 76.99(+1.03) | 69.85(+0.46) | 69.37(+0.65) | 77.50(+1.24) | 69.22(+0.41) |

Table 2: Applying our DMTD achieves new SOTA distillation results on Tiny-ImageNet. KD, DKD, and NKD all benefit from our DMTD. Notably, for the ResNet50-MobileNetV1 pair on Tiny-ImageNet, DMT-DKD achieves a significant +1.97% accuracy gain.

| | Teacher | Student | KD | **DMT-KD** | DKD | **DMT-DKD** | NKD | **DMT-NKD** |
|---|---|---|---|---|---|---|---|---|
| | | | *ResNet-34 as the teacher, ResNet-18 as the student* | | | | | |
| Top1 | 74.25 | 70.64 | 71.52 | **72.33(+0.81)** | 72.68 | **72.92(+0.24)** | 72.80 | **73.22(+0.42)** |
| Top5 | 92.36 | 89.93 | 90.77 | 90.93 | 91.31 | 92.18 | 92.09 | **92.25** |
| | | | *ResNet-50 as the teacher, MobileNet-V1 as the student* | | | | | |
| Top1 | 77.06 | 67.97 | 71.40 | **72.31(+0.91)** | 71.15 | **73.12(+1.97)** | 73.48 | **74.03(+0.55)** |
| Top5 | 93.76 | 87.86 | 91.24 | 91.64 | 90.15 | 90.99 | 91.70 | 92.03 |

knowledge, making fine-grained classification tasks more dependent on local semantic knowledge than classical classification tasks. Meanwhile, when trained on this dataset, DMTD learned higher global and local temperatures compared to the previous two datasets, indicating that the student tend to imitate the predictive behavior made by the teacher when faced with challenging classification tasks. If a fixed lower temperature value is forcibly set (such as $\tau$=4), the ability gap between the teacher and the student may lead to poor distillation performance.

**MS-COCO Object Detection.** We also apply our method to the object detection task. We follow the implementation of DKD for object detection and integrate our DMTD with the conventional DKD for this task, achieving results that surpass feature distillation in some cases. As shown in Tab. 4, our DMTD can further enhance the detection performance.

Table 3: Performance on the CUB200 Dataset. Here, we conducted experiments on three different types of teacher-student pairs: same architecture and layer (VGG13-VGG8), different architecture but same layer (ResNet32×4-ShuffleNetV1 and VGG13-MobileNetV2), and different architecture and layer (ResNet32×4-MobileNetV2 and ResNet50-ShuffleNetV1).

| Distillation Manner | Teacher | ResNet32×4 | ResNet32×4 | VGG13 | VGG13 | ResNet50 |
|---|---|---|---|---|---|---|
| | Acc | 66.17 | 66.17 | 70.19 | 70.19 | 60.01 |
| | Student | MobileNetV2 | ShuffleNetV1 | MobileNetV2 | VGG8 | ShuffleNetV1 |
| | Acc | 40.23 | 37.28 | 40.23 | 46.32 | 37.28 |
| Feature-Based | SP (ICCV'19) | 48.49 | 61.83 | 44.28 | 54.78 | 55.31 |
| | CRD (ICLR'19) | 57.45 | 62.28 | 56.45 | 66.10 | 57.45 |
| | MGD (ECCV'22) | - | - | - | 66.89 | 57.12 |
| Logit-Based | KD (NIPS'14) | 56.09 | 61.68 | 53.98 | 64.18 | 57.21 |
| | SD-KD (CVPR'24) | 60.51 | 65.46 | **59.80** | **67.32** | 60.56 |
| | DMT-KD (ours) | **61.14(+5.05)** | **66.17(+4.49)** | 58.96(+4.98) | 66.93(+2.75) | **61.25(+4.04)** |
| | DKD (CVPR'22) | 59.94 | 64.51 | 58.45 | 67.20 | 59.21 |
| | SD-DKD (CVPR'24) | 62.97 | 65.58 | 64.86 | **68.67** | 60.66 |
| | DMT-DKD (ours) | **65.15(+5.21)** | **67.45(+2.94)** | **66.48(+8.03)** | 68.14(+0.94) | **61.08(+1.27)** |
| | NKD (ICCV'23) | 59.81 | 64.00 | 58.40 | 67.16 | 59.11 |
| | SD-NKD (CVPR'24) | **62.69** | 65.50 | **64.63** | **68.37** | **60.42** |
| | DMT-NKD (ours) | 61.82(+2.01) | **66.47(+2.47)** | 63.74(+5.34) | 67.41(+0.25) | 59.58(+0.47) |

Table 4: MS-COCO results based on Faster R-CNN (Ren, 2015) - FPN (Lin et al., 2017): AP evaluation on val2017.

|  | AP | AP50 | AP75 |
|---|---|---|---|
| T: RN-101 | 42.04 | 62.48 | 45.88 |
| S: RN-18 | 33.26 | 53.61 | 35.26 |
| DKD | 35.05 | 56.60 | 37.54 |
| **DMT-DKD** | **35.94** | **57.17** | **38.20** |
| T: RN-50 | 40.22 | 61.02 | 43.81 |
| S: MN-V2 | 29.47 | 48.87 | 30.90 |
| DKD | 32.34 | 53.77 | 34.01 |
| **DMT-DKD** | **33.86** | **54.91** | **35.75** |

Table 5: Ablation of various setting and component in DMT-DKD. We trained the student ShuffleNetV1 with a teacher ResNet-32×4 using 1×schedule and applied distillation on CIFAR-100. The baseline DKD accuracy is 76.45 on CIFAR-100 and 64.51 on CUB200.

| Ablation Terms | | Acc | |
|---|---|---|---|
| Multi-Temperature | Dynamic Temperature | CIFAR-100 | CUB200 |
| ✗ | ✗ | 76.45 | 64.51 |
| ✗ | ✓ | 76.93 | 65.09 |
| ✓ | ✗ | 77.30 | 65.58 |
| ✓ | ✓ | **77.68** | **67.45** |

## 4.3 ABLATION STUDY

In the following experiments, we evaluated the effectiveness of multi-temperature setting and dynamic temperature component on CIFAR-100 and CUB200. We set ResNet-32×4 as the teacher model and ShuffleNet-V1 as the student model.

**Effect of the dynamic temperature module.** As shown in Tab. 5, incorporating the dynamic temperature module alone, without splitting the final feature maps, can also bring a slight performance gain. We believe this is mainly due to the adaptability of the dynamic temperature to the complexity of different samples. The result is 0.48 and 0.58 higher than the baseline respectively, indicating that the dynamic temperature module is important in DMT-DKD.

**Effect of setting multiple temperatures.** As shown in Tab. 5, we similarly split the final logit outputs by the teacher and student models according to the same scales as in the aforementioned DMT-DKD. Compared to using the same temperature $\tau=4$ for the logits at each scale, distillation with a higher temperature (e.g., 15) on global logits and lower temperatures (e.g., 14 and 4) on logits with $M=4$ and 16 respectively can achieve performance gains. This improvement arises because the model can fully leverage the advantages of feature maps at different scales. When processing global logits, the student tends to mimic the teacher's behavior when facing challenging samples with a higher temperature value, and when processing local logits, the student pays more attention to specific results, tending to mimic the teacher's ability to use key local features for correct inference with a lower temperature value. Furthermore, when the model is able to view data at multiple scales with different temperatures, it is forced to learn how to generalize its predictions across different circumstances, thereby enhancing its adaptability to unseen samples.

Compared to the DKD baseline, our DMT-DKD, with all settings and components incorporated, achieves a significant improvement of 2.24% (77.68% vs. 76.45%) on CIFAR-100 and 2.94% (67.45& vs. 64.51%) on CUB200.

## 4.4 FURTHER ANALYSIS

In this section, we first investigate the learning curve of temperature during the training process and then present visualizations from two perspectives (with the teacher set as ResNet-32×4 and the student set as ShuffleNet-V1 on CIFAR-100) for intuitive understanding.

**Learning curves of temperature during the training process.** Fig. 3 illustrates the learning curves of the temperature at different scales during the training process. The results validate our hypothesis. At the end of the training, the global logits converge to a higher temperature (approximately 9), while the local logits with $M=4$ converge to a lower temperature (approximately 3). This shows that the student model can adaptively adjust its learning strategy according to the feature scale, the global feature retains ambiguity and semantic information through high temperature, and the local feature focuses on key details through low temperature. This dynamic multi temperature mechanism enables the student model to make full use of the advantages of logits with different scales, so as to improve the overall performance.

**Inter-Class Distance Plot.** As illustrated in Fig. 4, the t-SNE visualization demonstrates that DMT-DKD achieves superior feature discriminability compared to DKD, with more compact intra-class

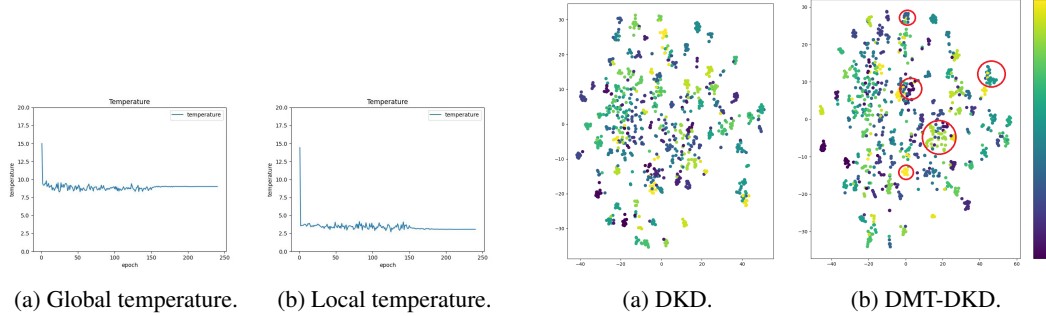

(a) Global temperature.  (b) Local temperature.

Figure 3: Learning curve of temperature during training. The model fully leverages the advantages of feature maps at different scales.

(a) DKD.  (b) DMT-DKD.

Figure 4: t-SNE of features learned by DKD and DMT-DKD.

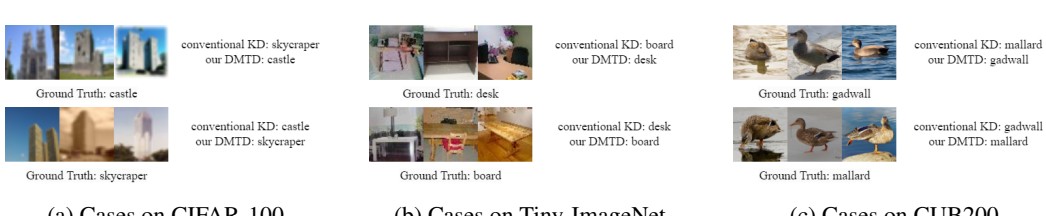

(a) Cases on CIFAR-100.  (b) Cases on Tiny-ImageNet.  (c) Cases on CUB200.

Figure 5: Some examples that students trained with DMTD can classify correctly, but those trained with conventional KD would misclassify.

clusters and clearer inter-class separation. This empirically validates that the proposed dynamic multi-temperature mechanism enables the student model to leverage both global semantic information (via higher temperatures) and local discriminative features (via lower temperatures), thereby enhancing the overall distillation performance.

Furthermore, we visualize some cases where students trained with DMTD can classify correctly while those trained with conventional KD make mistakes (Fig. 5). For instance, as depicted in Figure 5(c), on the CUB200 dataset, conventional Knowledge Distillation (KD) employs a single fixed temperature. This approach may lead to misclassification of gadwall and mallard, as these two bird species share similar global features, such as body shape and color. In contrast, our Dynamic Multi-Temperature Distillation (DMTD) method processes both global and local features through various dynamic temperatures. This not only preserves the ambiguity inherent in similar birds but also identifies crucial local details, such as head color and body markings. As a result, DMTD is able to accurately classify these birds. These results validate our proposition that DMTD can help student imitate both the inference results obtained by teachers based on local key features and the predictive behavior based on global integrated features.

## 5 CONCLUSION

This paper points out the limitations of the conventional logit distillation methods based on a single global temperature, which performs poorly when dealing with challenging computer vision tasks. To overcome these limitations, we propose a novel plug-in approach, namely Dynamic Multi-Temperature Distillation (DMTD), based on multiple learnable temperatures. This method enables the model to dynamically adjust the temperatures at various scales according to the significance of global and local features, enhancing the student model's ability to mimic the hidden behaviors of the teacher during inference. Experiments demonstrate that our DMTD can effectively integrate with representative logit distillation methods and achieve significant improvements across a wide range of teacher-student model pairs on multiple image classification and object detection benchmarks.

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

# A APPENDIX

## A.1 GENAI USAGE DISCLOSURE

We merely utilized ChatGPT during the preparation of this paper to assist with translation and polishing. Apart from this, we have not used GenAI to assist with any other work. The GenAI-generated content was reviewed and edited by us to ensure accuracy and adherence to scholarly standards. The final work reflects the our original ideas and interpretations, with GenAI contributions limited to some non-critical sentences.

