# OpenReview forum: "DMTD: Dynamic Multi-Temperature Distillation"
_ICLR.cc/2026/Conference — ICLR 2026 Conference Withdrawn Submission_

### Official Review · Reviewer_9eLw · 2025-10-28

**Soundness:** 2
**Presentation:** 2
**Contribution:** 2
**Rating:** 2
**Confidence:** 4

**Summary:**

The paper proposes a distillation method dubbed DMTD, which learns multiple temperature parameters for scaling distributions resulting from different scales of feature maps. Experimental results on image classification and object detection are provided to demonstrated the effectiveness of DMTD integrated to several logit distillation methods.

**Strengths:**

1. The proposed method enjoys a plug-and-play nature.
2. The proposed method is simple and computationally efficient.

**Weaknesses:**

1. The motivation of distillation with distributions obtained by multi-scale feature maps is not strong.
2. The proposed method lacks theoretical grounding.
3. Experiments are not comprehensive enough. (1) Results on ImageNet are not provided. (2) All teacher-student pairs evaluated on CIFAR-100 are cross-architecture ones, while same-architecture ones are missing. (3) No results on transformer-based models, e.g., ViT, are presented.
4. The analysis conducted in the paper doesn't provide much insight.

Minor weakness:
1. M in Eq. 7 should depend on n, and w should depend on both m and n.

**Questions:**

1. Which output distribution is used for student's final prediction? Is it the global one or a mixture of global and local ones (e.g., through averaging or majority vote)? If it's the former, then why are the local ones ignored during inference?
2. Temperature is trained to maximize the loss instead of minimizing. Why's that? This counterintuitive choice needs justification. Would training temperatures to minimize the loss lead to better performance?
3. $\beta=2>1$. Why do the samples that the teacher predicts wrongly get a larger weight? This counterintuitive choice needs justification.
4. What measures are taken if the sizes of feature maps cannot be split into M non-overlapping patches (e.g., a 7x7 feature map cannot be split into 4 non-overlapping patches)? Padding or overlapping the patches? Also, what should be done if the model is a ViT where features are not arranged in 2D?

---

### Official Review · Reviewer_Crcu · 2025-10-29

**Soundness:** 2
**Presentation:** 2
**Contribution:** 1
**Rating:** 2
**Confidence:** 4

**Summary:**

This paper proposes a knowledge distillation (KD) method based on multi-scale logit maps with trainable temperatures. In the method, multiple logit maps are constructed by applying local average pooling of diverse scales to feature maps, and then are respectively fed into knowledge distillation losses. The multi-scale knowledge distillations are equipped with distinct temperatures which are trained in an adversarial manner. In the experiments on image classification and detection tasks using benchmark datasets, the proposed method produces competitive performance to the other logit-based KD approaches.

**Strengths:**

+ Multi-scale logit maps are effectively leveraged to boost KD performance in comparison to the standard vector-based (global) logits.
+ Dynamic temperature learning aligns softmax temperatures to respective logit maps by exploiting diverse characteristics of multi-scale representation in an adversarial way.

**Weaknesses:**

- This paper misses describing the fundamental process to construct logit maps. Most deep models, especially the ones used in Sec.4, produce a $C$-dimensional logits (vector) to cope with $C$-class classification. The logit map could be obtained only in the case that global average pooling (GAP) summarizes a feature map into a feature vector which is further converted to the $C$-dim logit vector via FC layer, as shown in [R1]. So, it contradicts the authors' claim that the proposed approach can be plugged in any deep models; that is, the method is applicable only to the model which contains GAP and FC layer.

- Therefore, one can say that the logit-map representation has been proposed in CAT-KD [R1] which minimizes discrepancy between the logit maps of student and teacher models in a similar way to this work. So, this paper lacks comparison to the closely related work [R1] in terms both of theoretical formulation and empirical performance; the performance scores reported in [R1] seems to be superior to those shown in Sec.4.2, while empirically analyzing scales of the logit maps in the experiments.

- The paper lacks clear rationale for applying adversarial learning to optimize the temperature (Sec.3.3). There is no analysis nor discussion about why the adversarial learning is effective and necessary for the dynamic temperature; it is important to show how and why the temperature can be converged to favorable one through the adversarial process.

- Similarly to the dynamic temperature, MLKD [R2] incorporates diversity of temperature into the KD framework by means of multiple temperatures. The MLKD also considers multi-level representations of logits to produce competitive performances to those shown in Sec.4; this paper lacks comparison to the work [R2].

- It is unclear how to determine the initial temperature for the temperature learning. Otherwise, the authors should show robustness of the adversarial temperature learning against the initialization.

- In the experimental results, it is necessary to show the learned (optimized) temperature values on all the KD settings to demonstrate adaptability of the dynamic temperature learning.

- In Fig.4, it is hard to find out the difference between those two distributions of (a) DKD and (b) DMT-DKD. And, what does the color indicate?

>[R1] Ziyao Guo et al. Class Attention Transfer Based Knowledge Distillation. In CVPR2023.
>
>[R2] Ying Jin et al. Multi-level Logit Distillation. In CVPR2023.


**Minor comments:**
- Presentation in Sec.3 is confusing.
	- Notations are less clear; at least, show dimensionality such as for $f$ and use $M_n$ instead of $M$ for the $n$-th logit map.
	- In line 213, what is "complete feature map"?
	- In line 219, $n\in N \rightarrow n \in \{1,\cdots,N\}$.
	- In line 241, what does weight $\lambda$ mean? Is that a learning rate for the temperature?
- In line 296, what is "weight of the distillation loss"? Does it mean $\alpha$ in Eq.8?
- It is better to evaluate performance on large-scale dataset such as ImageNet, as done in the other KD papers.

**Questions:**

See the weakness section.

---

### Official Review · Reviewer_vGvY · 2025-10-31

**Soundness:** 2
**Presentation:** 2
**Contribution:** 2
**Rating:** 2
**Confidence:** 4

**Summary:**

The paper proposes DMTD, a plug-in module that produces multiple, learnable temperatures applied to logits pooled at different spatial scales. The system splits final feature maps into multiple patch sets, pools them to get local logits, and learns a temperature per scale to shape the softmax distributions for logit-based KD. Experiments on CIFAR-100, Tiny-ImageNet, CUB200 and MS-COCO show consistent improvements when DMTD augments KD/DKD/NKD

**Strengths:**

1. The motivation is clear and correct. A single global temperature can miss local discriminative cues in vision tasks, especially fine-grained ones.
2. The problem genuinely exists in practice. The paper shows cases (CUB200, Tiny-ImageNet) where local detail matters and a single τ hurts performance.
3. The method is simple and practical. It’s a plug-in that works with existing logit-based KD losses, so adopters can try it without redesigning pipelines. The implementation overhead is modest.

**Weaknesses:**

**Major Issues**
1. Several prior works learn temperatures or use multi-scale signals (MKD, KD-fn, DKD, NKD and recent scale/region-based distillations are already related). The manuscript must more strongly delineate what is new vs. what prior pieces already provide.

2. Theoretical justification is weak. The paper presents intuitions and curves, but lacks a principled argument for why multiple temperatures per scale provably improve KL alignment or generalization. A short formal or empirical analysis would help.

3. Baselines are incomplete. Comparison set omits several recent logit-focused and temperature-aware baselines (the paper cites some related works but doesn’t test against all strong 2023–2025 logit methods). Hard to know if gains are from multi-scale pooling or from extra parameters.

4. No ImageNet or large-scale classification results are shown. Gains on CIFAR/Tiny/CUB are promising but may not transfer to full ImageNet or modern benchmarks.

5. Domain generalization not tested. All experiments are vision-only (classification + detection). It’s unclear if the idea helps NLP, speech, or multimodal KD where logits and local structure differ.

6. Ablation depth is shallow. Important hyperchoices (number of splits M, β defaulting to 2, curriculum schedule, choice of pooling) need more ablations to show robustness. Some claims (e.g., specific temperature ranges and curriculum) lack causal evidence.

**Minor Notes**

1. The table captions and reference formatting have small inconsistencies. For example, RLD is listed as ICCV’25 but cited as arXiv 2024. Fix dates/venues for clarity.

**Questions:**

1. Can you clarify precisely what conceptual or algorithmic component makes DMTD distinct? For instance, is the novelty in using learnable, per-scale temperatures that evolve dynamically, or in how local logits are generated?
2. Did you match the student’s parameter count against a KD baseline with a similar-sized head or auxiliary parameters to ensure improvements aren’t from added capacity alone?
3. Can you provide some reasoning or derivation (even simplified) explaining why varying τ across scales improves KL alignment or convergence?

---

### Official Review · Reviewer_ADi9 · 2025-11-01

**Soundness:** 2
**Presentation:** 2
**Contribution:** 2
**Rating:** 4
**Confidence:** 4

**Summary:**

The paper introduces Dynamic Multi-Temperature Distillation (DMTD), a method for knowledge distillation that addresses the limitations of traditional logit-based distillation, which typically uses a fixed temperature that only captures global features. DMTD uses multiple learnable temperatures that adaptively adjust based on the importance of both global and local features. This enhances the student model’s ability to replicate the teacher's behavior during inference. Experimental results show that DMTD works effectively with existing logit distillation methods, achieving notable improvements in image classification and object detection tasks.

**Strengths:**

1. The paper introduces Dynamic Multi-Temperature Distillation (DMTD), which addresses the limitations of existing single-temperature logit distillation methods.

2. DMTD is designed to be easily integrated with existing logit-based distillation methods. This makes it highly adaptable to current distillation frameworks, which could lead to wider adoption and application across various models.

3. The experiments demonstrate significant improvements across a variety of datasets (e.g., CIFAR-100, Tiny-ImageNet, CUB200) and tasks (image classification and object detection). This showcases the effectiveness of DMTD in improving student model performance.

**Weaknesses:**

1. The introduction of dynamic multi-temperature distillation and the associated logit splitting module adds complexity to the implementation. This could make it challenging for practitioners to adopt the method without significant adjustments to existing models. Besides, the added complexity of learnable temperatures and multi-scale feature maps could lead to increased computational overhead during training, particularly in larger models or datasets. This may hinder scalability in resource-constrained environments. The authors had better discuss the issue of additional complexity and runtime.

2. While the paper compares DMTD to several logit-based distillation methods, it does not explore other prominent feature-based distillation methods in depth. Additionally, some important logit-based KD methods are neglected, such as [1,2]. A more balanced comparison could provide a clearer picture of DMTD's relative advantages.

[1] Logit standardization in knowledge distillation, CVPR24.
[2] Scaled decoupled distillation, CVPR 24.

3. The paper mentions the use of various hyperparameters, such as temperature, α, β, but it does not go into sufficient detail about their impact across different datasets or models. More guidance on tuning these hyperparameters could be helpful for those attempting to apply DMTD in different contexts.

4. The paper only provides a the formulation of the proposed DMTD, while a well-explained theoretical basis for why the method helps address the claimed problem is absent.

**Questions:**

How does the DMTD method compare to other existing logit-based distillation techniques in terms of computational efficiency, especially when applied to large-scale datasets?

---

### Note · Authors · 2025-12-01

I have read and agree with the venue's withdrawal policy on behalf of myself and my co-authors.